Biological invasions; Bayesian; Decision making; Detection; Probability of Absence; Surveillance

**Author for correspondence:**
David S. L. Ramsey,
Email: david.ramsey@delwp.vic.gov.au

# Invasive species eradication: How do we declare success?

David S. L. Ramsey[1] [iD], Dean P. Anderson[2] and Andrew M. Gormley[2]

[1]Arthur Rylah Institute, Department of Environment, Land, Water and Planning, Heidelberg, VIC, Australia and [2]Manaaki Whenua – Landcare Research, Lincoln, New Zealand

## Abstract

Deciding whether or not eradication of an invasive species has been successful is one of the main dilemmas facing managers of eradication programmes. When the species is no longer being detected, a decision must be made about when to stop the eradication programme and declare success. In practice, this decision is usually based on ad hoc rules, which may be inefficient. Since surveillance undertaken to confirm species absence is imperfect, any declaration of eradication success must consider the risk and the consequences of being wrong. If surveillance is insufficient, then eradication may be falsely declared (a Type I error), whereas continuation of surveillance when eradication has already occurred wastes resources (a Type II error). We review the various methods that have been developed for quantifying these errors and incorporating them into the decision-making process. We conclude with an overview of future developments likely to improve the practice of determining invasive species eradication success.

## Impact statement

We review the latest quantitative methods that can be used to analyse surveillance data to estimate the probability of species absence, when no individuals are detected. These methods allow defendable and transparent decisions to be made about the probability of successful eradication. Decisions associated with eradication operations need to be evidence-based to ensure that cost-efficient strategies are adopted and to satisfy concerns of funders, policymakers, managers and the public.

## Introduction

The impacts of invasive species on ecosystems are becoming increasingly pervasive, threatening biodiversity, ecosystem functioning, agricultural productivity and human health (Myers et al., 2000; Simberloff, 2014; Seebens et al., 2018; Blackburn et al., 2019; Seebens et al., 2021). Early intervention against incursions of invasive species that aims for eradication represents some of the highest benefit/cost ratios for investments in biosecurity policy (Baxter et al., 2008). However, eradication of invasive species can be challenging, especially once the species has become established. Eradicating a pest species from an area requires removing all individuals and simultaneously preventing reinvasion (Bomford and O'Brien, 1995). Despite these challenges, the list of international eradications is growing rapidly and encompasses diverse taxa, with over 1,550 eradication events recorded in the Database of Island Invasive Species Eradications, including 1,081 successful eradications of 59 species (Spatz et al., 2022). New technologies and evidence-based strategies (Nugent et al., 2018; Murphy et al., 2019) are enabling eradication of pest species from increasingly larger islands and continental areas (Cruz et al., 2005; Carrion et al., 2011; Anderson et al., 2022a). With eradication programmes becoming more ambitious and logistically difficult, the need to provide evidenced-based criteria for evaluating the progress and success of eradication programmes is becoming more critical.

## Key questions

One of the main decisions facing managers attempting to eradicate an invasive species is deciding when eradication has occurred. Once the species is no longer being detected, a decision must be made about when to stop the eradication programme and declare success (Morrison et al., 2007; Ramsey et al., 2009). In many eradication programmes, this decision is based on ad hoc rules (Russell and Blackburn, 2017). One popular rule of thumb for declaring eradication success for animal pests is 2 years without a detection (e.g., Dominiak et al., 2011; Robinson and Copson, 2014; Russell et al., 2016), whereas 3–5 years without a detection is often used for weeds (Rejmanek and Pitcairn, 2002).





However, using ad hoc rules of thumb based on surveillance or waiting for arbitrary periods of time with no detections has several issues. The main difficulty is that the selected time to declare success may not be optimal. The optimal time for declaring eradication successful is one that takes into account the consequences of making an erroneous decision. If surveillance is insufficient, then eradication may be falsely declared, resulting in the population continuing to spread and cause negative impacts (a Type I error), whereas continuation of surveillance when eradication has already occurred wastes resources (a Type II error). Both of these types of errors incur costs, and the optimal decision is one that minimises these costs (Regan et al., 2006). Here, we review the various methods that have been developed for quantifying these errors and incorporating them into the decision-making process for declaring eradication success. A glossary of important terms is included (Table 1).

## Methods developed for examining eradication success

Collection of surveillance data to confirm eradication success is usually undertaken at the point when eradication is suspected to have occurred; hence, the data consist entirely (or almost entirely) of 'zeros' (absences). We define the period when active control of the species is being undertaken as the 'removal phase' and the period of surveillance to confirm eradication as the 'confirmation phase'. In the usual sequence of events, the confirmation phase only commences once individuals are no longer being detected. If surveillance detects the species of interest, clearly, the species has not been eradicated (although 'functional' eradication could still be claimed [Green and Grosholz, 2021]). However, when the surveillance data consist entirely of absence records, how confident can we be that eradication has occurred? Confidence in eradication can be quantified by the probability of eradication (or species absence). Hence, following a series of zero detections from surveillance activities, the primary quantity of interest is the probability of absence, given the species was not detected.

Occupancy models have been developed to estimate false negative errors in biological species surveys (the probability the species was present but was not detected) (MacKenzie et al., 2002; Tyre et al., 2003). These models also allow estimation of the complement, the probability of absence given no detections. Extensions have involved the development of dynamic occupancy models, which allow estimation of colonisation and extinction rates, in addition to site occupancy (MacKenzie et al., 2006). However, estimation of site occupancy requires the collection of spatially and temporally structured data on species presence and absence (e.g., using a sampling design), which can be labour-intensive and may not be possible towards the end of an eradication programme, when the species is mostly absent. In addition, the use of multiple types of monitoring data, both structured and unstructured (e.g., sighting reports collected haphazardly by the public), presents difficulties for use in occupancy models. Hence, using occupancy models to estimate eradication success may not always be practical or even feasible.

Several authors have used a time series of presence and absence records of a species (i.e., sighting records) to infer species absence (Solow, 1993; Solow et al., 2008; Rout et al., 2009a). Interest is usually focused on the tail of the record, when sightings are sparse, and the inference is based on the number of absent sighting occasions deemed necessary for declaring absence. These methods model the unknown observation process by assuming that the underlying sighting rate of the species is either constant or declining and follows a stationary or nonstationary Poisson process. Various modifications of this approach have been developed to allow flexibility in the sighting process specifications, for example, modifications for handling uncertain sightings (Lee, 2014), and for increasing or decreasing populations (Caley and Barry, 2014). However, the incorporation of structured and unstructured surveillance data presents difficulties for these methods, especially if surveillance effort is nonconstant in space or time.

Early work on inferring species eradication or extinction proposed using a null hypothesis testing framework to inform the decision about when to declare a species absent after a series of zero sightings (e.g., Solow, 1993; Reed, 1996; Solow and Roberts, 2003; McInerny et al., 2006). Hence, this approach addresses the question of how many zero sightings are probable, given the species is extant (null hypothesis), setting a threshold for this probability (Type I error). Solow (1993) also provided an alternative framework that calculated the probability that the species was extant, given a sighting record, using Bayes' theorem. This framework required construction of the prior probability that the species was extant and used Bayes factors to assess the degree of support for this probability (Solow, 1993; Rout et al., 2009a). Regan et al. (2006) first proposed explicit consideration of the costs of making a Type I error (false declaration of eradication) or a Type II error (surveillance continues when species has been eradicated), adopting a Bayesian framework for inference. These costs were considered jointly, and eradication was declared when the net expected costs (*NEC*) were minimised. Hence, the optimal time to declare eradication success was a trade-off between the cost of ongoing surveillance and the cost of making a false declaration of eradication. The main issue with this approach was that uncertainty was not incorporated into the estimates of the detection (likelihood) and persistence (prior) parameters required by the model; hence, decisions may not be robust to uncertainty (Rout et al., 2009b)

### *Surveillance sensitivity*

During the confirmation phase, the probability of absence can be derived from estimates of the surveillance sensitivity (*SSe*), the probability of detecting the species within a region of interest, given it is present at some predetermined level (i.e., the 'design prevalence' – see below) (Martin et al., 2007). The *SSe* is subtly different from the detection probability that is derived from models fitted to monitoring data (e.g., occupancy models), which only condition on presence in a sampling unit. The *SSe* for a region is usually constructed from the sensitivities of the various types of surveillance, which can be either structured or unstructured (Martin et al., 2007). Given a series of zero detections, the *SSe* quantifies the effectiveness of the search effort (the probability of detection given the design prevalence), but it is not, per se, an appropriate indicator of eradication success. The probability of absence given no detections from surveillance can be derived from the *SSe* using Bayes' theorem, which also requires consideration of the prior probability of absence (i.e., the probability of absence prior to the confirmation phase). Given an estimate of the *SSe* and the prior probability of absence (*Prior*), the probability of species absence (*PoA*) is given by

$$PoA = \frac{Specificity \times Prior}{Specificity \times Prior + (1 - SSe)(1 - Prior)}, \quad (1)$$

where *Specificity* is the probability of not detecting the species when the species is not present. Equation (1) is analogous to the

**Table 1.** Glossary of terms used in text, including abbreviations and brief definitions

| Terms | Abbreviations | Definitions |
|---|---|---|
| Probability of absence | PoA | Probability that the target species is absent from the area of interest, given no detections |
| Surveillance sensitivity | SSe | Probability of detecting an individual within the total area of interest, given that $P_u$ cells are occupied |
| Prior | Prior | The starting probability of absence (*PoA*) before surveillance has begun |
| Design prevalence | Pu | Statistical parameter defining the number of occupied cells in the surveillance data model |
| Maximum probability of detection | $g_0$ | Probability of detection during a single time interval for a device placed at the home range centre |
| Spatial detection decay parameter | $\sigma$ | Rate of decay in the probability of detection with increasing distance between the home range centre and the device |
| Type I error | | Falsely declaring a species eradicated (and ceasing surveillance) |
| Type II error | | Falsely declaring a species extant (and continuing surveillance) |
| Net expected cost | NEC | Joint expected cost of making a Type I or Type II error |
| Stopping rule | | Criteria used to determine when an eradication programme ceases |

negative predictive value of a diagnostic test used in disease surveillance (Martin et al., 2007). If we can assume that the *Specificity* is equal to 1.0 (i.e., no false positive detections), then equation (1) simplifies to

$$PoA = \frac{Prior}{1 - SSe(1 - Prior)}. \quad (2)$$

The *Prior* can be obtained in a number of ways, including (i) expert opinion (Ramsey et al., 2009), (ii) meta-analysis of eradication programmes from similar species (Dodd et al., 2015), or (iii) use of models to simulate lethal control (Gormley et al., 2016).

The *PoA* is the metric used to guide decisions, which incorporates the *Prior* and the *SSe*. The following hypothetical example illustrates the importance of the *Prior* and why we bother with Bayesian logic. Consider two identical islands on which toxic baits were used to remove rats (Samaniego-Herrera et al., 2013). The first island had complete bait coverage, whereas the second had large gaps in bait deployment. The higher operational investment on the first island results in a higher prior probability of success (before the confirmation phase) than on the second island. The subsequent surveillance during the confirmation phase was equal on both islands (i.e., equal *SSe*), and no rats were detected. Combining the *Prior* with the *SSe* demonstrates, intuitively and quantitatively, that confidence in eradication success is higher for the first island than for the second.

The value specified for the *Prior* has a large influence on the level of surveillance that needs to be conducted to confidently declare absence of the pest (Figure 1A). For example, if we have low confidence that control was sufficient to eradicate the pest (*Prior* = 0.5), then surveillance efforts need to be extremely high (*SSe* = 0.9) to achieve a high level of confidence in successful eradication (*PoA* > 90%). Conversely, if the *Prior* = 0.8, then surveillance efforts can be reduced (yielding an *SSe* = 0.6) to achieve the same level of confidence regarding absence of the pest (*PoA* > 90%).

Quantitative planning increases the chances that a cost-effective surveillance strategy will be deployed (Gormley et al., 2018). We can rearrange equation (2) to determine the level of surveillance required ($SSe_{req}$) to improve our level of confidence in eradication from the *Prior* to the target *PoA* ($PoA_{Target}$):

$$SSe_{req} = \frac{PoA_{Target} - Prior}{PoA_{Target}(1 - Prior)}. \quad (3)$$

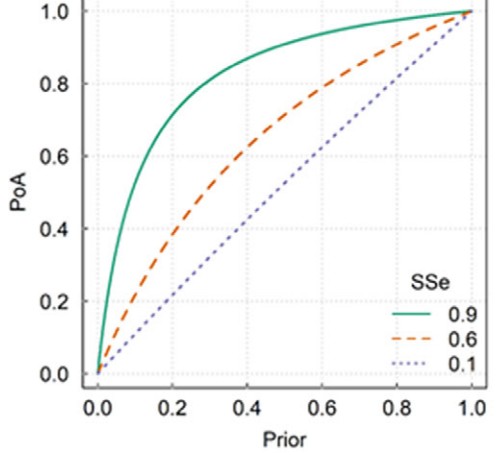
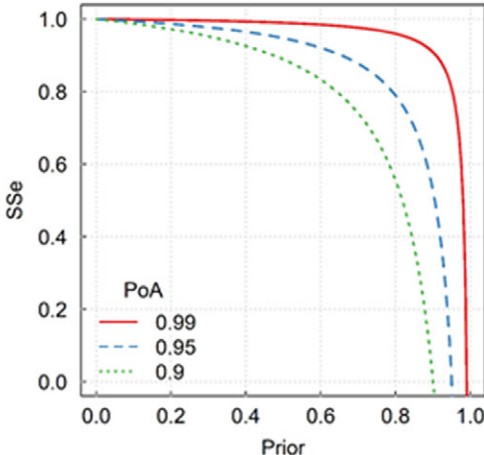

**Figure 1.** (A) Contour plot showing the relationship between the starting probability of absence (*Prior*) (*x*-axis) and the resulting probability of absence (*PoA*) (*y*-axis) for three levels of surveillance sensitivities (*SSe*) (contour lines). (B) Contour plot showing the relationship between the *Prior* (*x*-axis), the *PoA* (contour lines) and the *SSe* (*y*-axis).

For example, if the $Prior = 0.9$ (i.e., we are 90% sure of eradication after control) and $PoA_{\text{Target}} = 0.95$ (i.e., we want to be 95% sure of success), then $SSe_{\text{req}} = 0.53$; this means that we need to do enough surveillance to have a 53% chance of detecting any remaining individuals at the design prevalence (Figure 1B). If, however, we wanted to be 99% sure of success, then for the same $Prior$, a much higher level of surveillance would be needed (i.e., $SSe_{\text{req}} = 0.91$).

### Spatial PoA model

Methods have been developed to estimate $SSe$ by incorporating information on the spatial deployment of monitoring devices across the area of interest, and on attributes of the target species (Anderson et al., 2013; Kim et al., 2020). The surveillance model is based on a simple spatial model for the detection of individuals based on a function of the distance between an individual and a detection device. Individuals are assumed to occupy a symmetric home range, and detection declines with increasing distance between the home range centre and the device location. This spatial detection process is governed by two parameters: $g_0$ – the probability of detection over a single time interval by a device placed at the home range centre (i.e., the maximum probability of detection); and $\sigma$ – the rate of decay in the probability of detection with increasing distance between the home range centre and the device (Efford, 2004). For simplicity, a half-normal function is usually used to model the decay in detection probability, with $\sigma$ being equivalent to the standard deviation of the circular normal kernel. This parameter is proportional to the home range size of an individual. This simple model was first used to model detection in spatially explicit capture–recapture models (Efford, 2004; Borchers and Efford, 2008). However, this spatial detection function has also proved to be useful in simulation models for designing efficient surveillance for achieving management objectives (Ramsey et al., 2005; Gormley and Warburton, 2020; Anderson et al., 2022a).

The spatial surveillance approach is constructed by superimposing a spatially referenced grid-cell system on the area of interest (i.e., a raster layer). Each grid cell corresponds to a sampling unit, and the model quantifies the probability of detecting an individual in each grid cell, given a surviving individual's home range centre is located in the grid cell. Detection devices or search effort in or around a grid cell have a chance of detecting an individual. Each device type has its own maximum detection probability ($g_0$), with $\sigma$ derived from home range estimates for the species. A spatially explicit detection surface is quantified by adding detection kernels for each device location. The height (or intensity) of the kernel is equivalent to the amount of sampling effort undertaken by that device (e.g., number of trap nights). The number of grid cells covered by all kernels determines the proportion of the total area covered by surveillance. Alternatively, grid cells can be searched directly with the probability of detection related to the amount of search effort in a cell. Methods based on search effort by observers are most often employed during surveillance for weeds (e.g., Garrard et al., 2008; Hauser et al., 2022). The grid-cell approach allows the accommodation of diverse combinations of surveillance information, which might vary by detection method, location, sampling effort, and deployment period. The use of multiple detection methods, including those not requiring an interaction by the animal (such as a camera or eDNA), can improve the chances of detecting device-shy individuals.

The spatially explicit surveillance model also incorporates habitat selection by the target species because the likely location of a limited number of survivors is not expected to be equal across the landscape but concentrated in preferred areas. Resource selection studies (Manly et al., 2002) and the results from species distribution models (Elith et al., 2006) can inform the relative probabilities of survivors in different locations and assist in the creation of a relative-risk map (Anderson et al., 2013, 2022a). The resolution of the grid-cell system superimposed on the eradication area should be finer than the home range size and should also accommodate spatial heterogeneity of the relative-risk map. The estimated $SSe$ will be maximised when search effort is spatially distributed proportionate to the relative risk of survivor presence (Martin et al., 2007).

The $SSe$ for the eradication area is calculated by combining the spatial surveillance surface (grid-cell-level probabilities of detection), the relative-risk map of habitat use, and a statistical parameter representing the minimum number of occupied grid cells ($P_u$) that are available to be detected. The latter element is referred to as 'design prevalence' in disease surveillance (Cameron and Baldock, 1998) and determines the definition of the $SSe$. For example, if the minimum number of occupied grid cells is set to 1, the $SSe$ is defined as 'the probability of detecting an individual given that only one grid cell is occupied in the area of interest'. Intuitively, it is easier to detect one of many occupied grid cells than a single occupied grid cell. When aiming to confirm eradication success, we are trying to find the last survivor, or one of a few remaining survivors. Therefore, the minimum number of occupied grid cells is generally set to 1 (however, see 'Extensions to the $PoA$ model' below). If we obtain a high $SSe$ assuming only one occupied cell, and do not detect anything, we can increase our confidence that less than one cell remains occupied, that is, zero are present.

Values for the spatially explicit detection parameters ($g_0$ and $\sigma$) of animals in monitoring devices can be obtained from published reports, experimental or field studies (Efford, 2004; Ball et al., 2005; Ramsey et al., 2015; Anderson et al., 2022b) or expert opinion (Anderson et al., 2022a). Similarly, detection experiments have typically been used to estimate the probability of weed detection given a certain amount of search effort (Garrard et al., 2008; Hauser et al., 2022). These parameters are input into the model as distributions in order to account for uncertainty. High variances should be used where there is high parameter uncertainty, which is propagated through to estimates of $SSe$ (Anderson et al., 2022a). Given high parameter uncertainty, increasing sample size, through increased surveillance effort, will increase the mean and decrease the variance of the $SSe$.

Once an estimate of $SSe$ and its uncertainty is obtained, the $PoA$ (and associated variance) can be calculated, initially by updating the $Prior$ (equation (2)). The resulting $PoA$ then becomes the prior probability for the next round of surveillance data, and so forth. This updating of the $PoA$ continues as new surveillance data are added (Anderson et al., 2013; Ramsey et al., 2022), until the mean $PoA$ exceeds a target threshold, which is set by stopping rules (see below).

### Stopping rules

No matter how much surveillance is undertaken, managers can never be certain about eradication success, due to uncertainty in the detection process. Decisions on when to declare success must consider the risk of being wrong. As stated above, this risk can be encapsulated by the Type I and Type II error rates and the consequences of making a wrong decision. A stopping rule is a statement

about the criteria for ceasing an eradication programme, which may or may not consider these error rates.

Intuitively, successful eradication can be declared when there is a high probability that the residual population is zero. This is equivalent to minimising the Type I error rate, the probability that eradication is wrongly declared. A logical stopping rule would involve a threshold for the *PoA* that, when exceeded, triggers declaration of eradication success. Typically, thresholds are set such that eradication success is declared once the *PoA* exceeds some value, such as 95% or 99% (e.g., Ramsey et al., 2009, 2011; Anderson et al., 2017). A stopping rule using a 95% threshold for the probability of absence is equivalent to saying that one out of 20 similar eradication attempts with equivalent effort would fail to detect survivors. The advantage of this type of stopping rule is its relative transparency; the level of certainty is clear to managers. The disadvantage of this type of stopping rule is that picking a threshold for the Type I error rate is arbitrary.

A second type of stopping rule considers both the Type I and Type II error rates, by examining the joint costs associated with these errors. These costs can be summarised as the cost of surveillance plus the expected cost that would be incurred if eradication were to be wrongly declared. The optimal time for declaring eradication successful is when the *NEC* is minimised (Regan et al., 2006). A stopping rule based on minimising the *NEC* avoids the issue of setting a threshold for the *PoA*. While theoretically sound, implementing this stopping rule has practical difficulties. The main difficulty is that the expected cost of wrongly declaring successful eradication is not easily quantified. This is because these costs include both tangible costs (e.g., the cost of repeating the eradication attempt) and intangible costs (e.g., reputational costs or biodiversity loss associated with the failure to eradicate). In many eradication attempts, the intangible costs are deemed to be high, but they are difficult or even impossible to quantify (e.g., costs due to biodiversity loss). Many managers are primarily concerned with the intangible costs and thus try to minimise the Type I error. Recently, attempts have been made to address this through a utility function that considers both the cost variance and the expected costs, incorporating a parameter indicating the degree of 'risk aversion'. This is then used to optimise the most cost-effective threshold for the *PoA* (Gormley et al., 2018).

### Extensions to the PoA model

Extensions to the spatial *PoA* model have been developed, principally to allow the model to be applied to large eradication programmes (Anderson et al., 2017). Large (or broadscale) eradication programmes are defined as ones in which management of the species cannot occur concurrently over the entire area of interest. The eradications of Bovine Tb (*Mycobacterium bovis*) from wildlife in New Zealand (Livingstone et al., 2015), fire ants (*Solenopsis invicta*) from south-east Queensland, Australia (Spring and Cacho, 2015) and nutria (*Myocastor coypus*) from the Delmarva Peninsula, USA (Anderson et al., 2022a) are all attempting eradications over 0.5–2.0 M ha. By necessity, these large areas are often subdivided into smaller management zones, in each of which, eradication actions operate as a single unit and are large enough to minimise the risk of reinvasion from neighbouring zones. Eradication then proceeds in two stages. Stage I involves the removal of the species, followed by confirmation phase surveillance to declare absence within each zone. Once a management zone is declared free of the species in Stage I, it then passes to Stage II and the operational resources are reallocated to the next zone. In this way, eradication

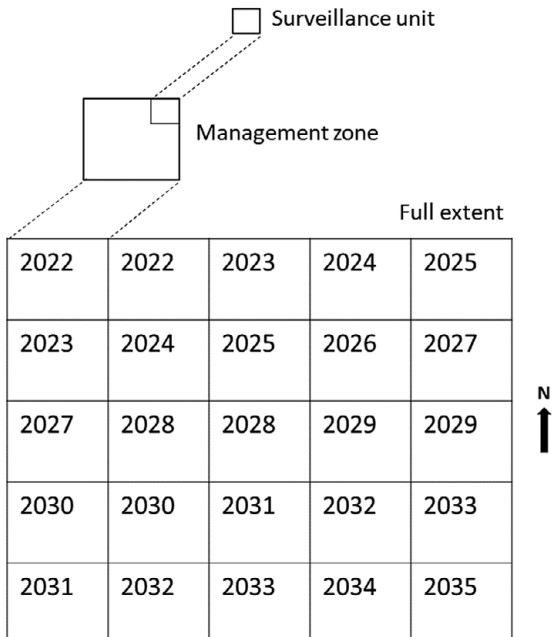

**Figure 2.** The spatiotemporal progression of a hypothetical broadscale eradication operation over a square-shaped region, which begins in the north-west of the region in 2022 and finishes in the south-east in 2035 (modified from Anderson et al., 2017). Each square represents a management zone for control purposes, and the number in each represents the year in which control is to be undertaken in the zone. Surveillance devices or search efforts are allocated to the surveillance unit (smallest squares, top of figure). Stage I is the period in which control is being undertaken, and 'freedom' is the criterion required for an operational decision at the management-zone level to allow reallocation of resources to other management zones, that is, progression of the operation across the landscape. Stage II entails ongoing surveillance in management zones declared 'free' at the end of Stage I. The purpose of Stage II is to identify erroneous freedom declarations, and to eventually declare the species eradicated from the entire broadscale area. Confirmation of eradication in Stage II may extend well beyond 2035.

proceeds progressively over the entire extent until all zones are declared free of the species (Figure 2). Importantly, once a zone progresses to Stage II, surveillance in that zone should continue, so that any residual survivors or incursions are detected. Since the majority of resources are committed to zones undergoing Stage I, Stage II surveillance data sources will usually be low-cost/low-intensity sources, such as reports from the public or other passive surveillance sources.

An important point is that all zones being declared free of the species at Stage I does not necessarily equate to a high level of confidence in eradication over the entire extent. Consider 10 management zones declared free using a 95% threshold for the *PoA*. Hence, each zone has a Type I error rate of 5% of being incorrectly declared free, and therefore the probability that at least one of the 10 zones has been incorrectly declared free is $1 - (1 - 0.05)^{10} = 0.4$, giving a *PoA* over the entire extent of 0.6. To achieve high confidence in eradication over the entire extent, Stage II surveillance must be used. Since the management zones have been undergoing Stage II surveillance for various periods of time (i.e., since declaration of absence of the species at the end of Stage I), calculation of the *SSe* for each zone needs to incorporate this variable time under surveillance. This is achieved by assuming that the residual population (if present) should increase within the zone with the passage of time. Under positive population growth, detection of a species should become more likely over time due to population increase

and spread. This is reflected in the calculations of the *SSe* for each management zone by allowing the minimum number of occupied cells ($P_u$) to increase over the period of Stage II surveillance. This can be achieved, for example, by assuming that $P_u$ increases according to the logistic growth function

$$P_{u(t)} = P_{u(t-1)} + rP_{u(t-1)}\left[1 - P_{u(t-1)}/K\right], \qquad (4)$$

where $r$ equals the intrinsic growth rate and $K$ is the carrying capacity. Assuming $K$ is large relative to population size (as is expected in a population of residual survivors), equation (4) can be approximated by

$$P_{u(t)} = P_{u(t-1)}(1+r). \qquad (5)$$

Allowing $P_u$ to increase due to equation (5) means that, even if the *SSe* is initially low, it will increase over time because undetected survivors would be expected to increase, making them easier to detect (Caley et al., 2015; Anderson et al., 2017).

### Approximations to the spatial PoA model

The spatial *PoA* model outlined above represents a flexible and powerful tool for quantifying eradication success. However, there are several limitations. Calculations of the uncertainty in the estimates of the *SSe* and the *PoA* are derived from Monte Carlo simulations based on the underlying probability distributions of the component parts (e.g., $g_0$, $\sigma$, and *Prior*). Usually, many draws are required to reduce Monte Carlo errors, so processing models utilising data from large extents over many years is computationally expensive. Recently, analytical Bayesian solutions to the *PoA* model have been developed (Barnes et al., 2021, 2022). The analytical solutions are based on probability-generating functions, which fully define discrete distributions (Feller, 1958). Using standard statistical theory, the stochastic processes in the *PoA* model can be expressed as compound distributions from which analytical solutions can be determined. These solutions can then provide a straightforward means of deriving posterior distributions and statistics (Barnes et al., 2021, 2022). One advantage of these analytical formulations is that they allow a more tractable analysis of surveillance design, making exploration of the cost of alternative strategies, the impacts of stochasticity and parameter uncertainty much more computationally efficient.

### Outlook and future directions

Declaring successful eradication of invasive species has come a long way from the use of simple ad hoc rules that rely on a 'wait-and-see' approach. The proof of absence framework enables calculation of the *PoA* using a wide variety of surveillance types. Using the power of Bayesian updating, managers can make informed, evidence-based decisions as to whether eradication can be declared with a degree of confidence or whether more surveillance is needed. In addition, methods now exist for quantitatively assessing surveillance strategies so as to ensure that the most cost-efficient strategies are adopted for declaring eradication success.

Challenges for the implementation of these surveillance models are finding efficient ways of obtaining the parameters of the component species-specific detection probabilities for each surveillance method ($g_0$), especially for novel monitoring techniques. For weed species, studies have demonstrated how detection probabilities can be related to species traits and observer experience (Garrard et al.,

2013) and similar trait-based models may be applicable to the detection parameters for animal species. One barrier to the uptake of analytical methods now available to managers of eradication programmes is their complexity: managers need to have some quantitative skills for their successful implementation. Current work that aims to deliver these models within a user-friendly computer programme or interface should greatly lower the barriers to their use, enabling managers to confidently determine the optimal amount of surveillance required to declare eradication, allowing more efficient use of resources.

**Open peer review.**   To view the open peer review materials for this article, please visit http://doi.org/10.1017/ext.2023.1.

**Data availability statement.**   No data were analysed in undertaking this review.

**Author contributions.**   D.S.L.R. and D.P.A. conceived the scope and outline of this review and led the writing of the manuscript with significant contributions by A.M.G. All authors contributed critically to drafts and approved the final version of the manuscript.

**Financial support.**   The authors received no financial support for the research and authorship of this manuscript.

**Competing interest.**   The authors report no competing interest.

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
