## [Reviewer Report]

*Comments to Author*: This paper presents a good and timely mini review of the growing research area of statistical confirmation of invasive species eradication. The paper is well written and technically correct, but I have some issue with framing of some aspects and absence of others which I outline below.

Mt first is that the authors argue that waiting two years is subjective, ad-hoc and unscientific, but alternatively, I simply consider this resource inefficient (aka Type II). By waiting two years (which is a rule that emerged from rodent eradications) one is really just waiting until one would be 100% confident that no detection can only be eradication (or conversely failure is readily detected with minimal effort). Alternatively, its just a risk averse form of NEC. I think in the discussion they make a fairer summary which is that this approach is simply ad hoc (but not necessarily subjective nor unscientific).

This leads to my second point which is rodent eradications (and others) are confirmed through surveillance independent and subsequent to the control, whereas throughout the manuscript the authors contextualise eradications where control and confirmation are simultaneous in their methodology (as they can be for larger mammal eradications). When eradication confirmation can be timed independently of the eradication operation, arguably waiting longer to do it is more cost effective (especially so if two years means you don’t even have to do any surveillance!). The only reason for doing more rapid independent confirmation assessment would be to confirm eradication earlier, for some reason (e.g. rodent eradication failure rapid responses are now starting to happen, or desire to reintroduce threatened species, etc).

This leads to my third point which is that from the original PoA work by Anderson and colleagues a spin-off focused on island rodent eradications has developed (rapid eradication assessment: REA www.rea.is – that has been available online for over 5 years with a user friendly interface as the authors themselves argue for). This isn’t referenced in the text, but the most recent paper on it contains some advances (e.g. coverage, incursion c.f. whole island eradication confirmations and the use of mobile detection devices a.k.a dogs but could be drones, etc). See: https://www.publish.csiro.au/WR/WR18154.

Given the authors consider the 2-year eradication confirmation period subjective I was surprised that a Bayesian approach was not given more attention as also being subjective (in its selection of priors). I certainly have no issue with the Bayesian framing for PoA, but in my own experience the PoA/REA result is incredibly sensitive (a.k.a. dependent upon) the prior (as the authors highlight themselves) and ultimately, the selection of the prior is subjective (although any given prior can be objectively informed). This requires discussing, and certainly requires users of PoA to perform sensitivity analysis to priors lest they don’t realise their PoA is simply their prior re-packaged.

Another issue with PoA/REA is distinguishing device specific variation on g0, from individual variation in animal behaviour. I’ve essentially landed on the g0 distribution being a compound distribution of these two, so mathematically it doesn’t matter, but ecologically I think there remains interest in which of these two contributes most to variation in modelled g0. From that then follows the role of animal personalities…if an animal is completely adverse to say trapping (e.g. recalcitrant https://www.sciencedirect.com/science/article/pii/S0169534720301877), then no amount of trapping will ever detect it. So multiple detection tools are required (as is standard in rodent incursion response). PoA/REA doesn’t currently capture this possibility.

Another issue in PoA/REA is coverage of devices by way of summed sigma footprint over the confirmation area. This is critical and an understanding of it helps frame interpretation of PoA/REA results.

For large landscape eradication projects where reinvasion is non-negligible, an ongoing issue in eradication confirmation is distinguishing eradication survivors from reinvaders. The longer one waits to confirm eradication the easier it is to confirm, but also the more likely it is to confound reinvaders from survivors.

Ultimately, with PoA/REA, I personally recommend it not as an objective quantitative result itself (for all the foibles I mention here), but as a tool which allows managers to query their own assumptions, e.g. coverage, desired confidence, etc…and decide if they (albeit ultimately still then subjectively) are confident in the declaration of eradication success, although I realise this position in quantitative PoA may not be appealing to others.

---

## [Reviewer Report]

*Comments to Author*: I have written a couple of reviews on methods for declaring eradication myself – one short conference proceeding paper in 2009 and a book chapter in 2017. References for both are below for full visibility. My reviews were targeted towards an audience of practitioners, so were perhaps not methodologically comprehensive but focused on talking through some worked examples of different types of methods. Given this I feel the current manuscript is distinct as it gives an updated review of methodology for a wider audience of scientists, analysts and practitioners.

I found the manuscript to be very well written and easy to read. I have a few comments below that relate mainly to how the methods have been conceptually categorised and how this has influenced the manuscript’s structure.

- Lines 70-79: does this also apply to dynamic occupancy models, which estimate time dependent rates of extinction/colonisation?

- Line 89: It would be good to include a brief definition and/or examples of structured and unstructured surveillance at first mention.

- Lines 91-101: it would be useful here (or generally in the structure of the manuscript) to draw a distinction between null hypothesis testing methods and Bayesian methods, because these methods are used differently when setting stopping rules.

The Reed 1996 paper mentioned here and most (but not all) of the sighting record literature (Boakes et al. 2015) test the null hypothesis that the species is extant, calculating the probability of obtaining observed data or more extreme data given that the species is extant (p-value). These methods can be used in stopping rules by setting a p-value at which to reject the null hypothesis, even setting it to minimise cost e.g., Field et al. 2004.

However, Regan et al.’s (2006) cost minimising stopping rule requires the Bayesian probability that the species is extant but undetected.

Some other notes on Bayesian versus null hypothesis testing approaches - I have found it is more natural for decision makers to consider their risk tolerance to the probability that the species remains extant given the data collected (Bayesian probability), rather than a p-value. The authors also mention an additional advantage of all Bayesian methods which is the ability to incorporate a prior probability that the species is extant.

- Lines 103-157: In some survey methods when a remaining individual is seen it is removed, and in some it is not always. Do the methods in this section cover both these cases? Is there a conceptual difference in the analysis methods available for projects where the population being eradicated is monitored during eradication efforts (e.g., collecting data on the number of foxes shot or weeds removed) versus a population where a control action is applied and then monitoring conducted afterwards (e.g., dropping rat baits on an island)? I have always thought that the inclusion of removal data opens up more options for modelling the population and calculating probability of eradication, e.g, Bayesian catch-effort models such as Ramsey et al. 2009 and Rout et al. 2014 and perhaps even dynamic occupancy models mentioned above.

References

Boakes et al. (2015) Inferring species extinction: the use of sighting records. Methods in Ecology and Evolution 6: 678-687.

Field et al. (2004) Minimizing the cost of environmental management decisions by optimizing statistical thresholds. Ecology Letters 7: 669-675.

Ramsey et al. (2009) Quantifying eradication success: the removal of feral pigs from Santa Cruz Island, California. Conservation Biology 23: 449-459.

Reed (1996) Using statistical probability to increase confidence of inferring species extinction. Conservation Biology 10(4): 1283-1285.

Regan et al. (2006) Optimal eradication: when to stop looking for an invasive plant. Ecology Letters 9: 759-766.

Rout (2009) Declaring eradication of invasive species: a review of methods for transparent decision-making. Plant Protection Quarterly 24: 92-94.

Rout et al. (2014) When to declare successful eradication of an invasive predator? Animal Conservation 17: 125-132.

Rout (2017) Declaring eradication of an invasive species, pp. 334-347 in Invasive Species: Risk Assessment and Management (ed Robinson AP, Walshe T, Burgman MA and Nunn M). Cambridge University Press, Cambridge UK.

---

## [Editor Report]

*Comments to Author*: Dear Dr. Ramsey

Your manuscript has now been reviewed by two referees, both of whom have expertise related to invasive species management, and in particular, knowledge pertaining to assessing the effectiveness of control methods. Both reviews comment that the manuscript is well written and quantitatively accurate, assessments that parallel my own impressions. Importantly, both reviewers comment that the manuscript is both timely and helpful as a review; these points are important given that the general subject matter of evaluating success of invasive species eradication methods has previously been reviewed to varying degrees in several contexts over the last few years.

Even though they are overall enthusiastic about the manuscripts' contributions, both reviewers have made several thoughtful suggestions about how the manuscript can be improved. In general, these comments will require only modest effort toward revision, and almost all of them can be accommodated only with some revised wording and discussion of additional relevant work.

One comment that I feel is especially important regards the relative timing of the eradication and assessment components of the management effort, with specific attention to whether these components are wholly consecutive, or whether there is some overlap in timing. This issue, which was discussed in similar language by both reviewers, parallels similar ideas discussed in the adaptive management literature, but here there are specific connections to the statistical modeling of assessment. As you and your colleagues undertake a revision, please pay close attention to the reviewers' comments in this area. 

I look forward to seeing your revision.

---

## [Reviewer Report]

*Comments to Author*: The authors have responded appropriately to all my previous comments, and their additions to the manuscript read well. I appreciate how they now draw the distinction between null hypothesis and Bayesian methods, and their explanation around focusing on methods for eradication confirmation as opposed to methods that model the control/eradication process – these additional explanations have more clearly outlined the scope of the review and how it relates to other similar literature. I’m happy for this version of the manuscript to be accepted for publication.

---

## [Editor Report]

*Comments to Author*: I concur with the reviewers that the authors have satisfactorily addressed the comments from the first review, and am happy to recommend acceptance